# StaR Is a Positive Regulator of Topoisomerase I Activity Involved in Supercoiling Maintenance in *Streptococcus pneumoniae*

**DOI:** 10.3390/ijms24065973

**Published:** 2023-03-22

**Authors:** Antonio A. de Vasconcelos Junior, Jose M. Tirado-Vélez, Antonio J. Martín-Galiano, Diego Megias, María-José Ferrándiz, Pablo Hernández, Mónica Amblar, Adela G. de la Campa

**Affiliations:** 1Centro Nacional de Microbiología, Instituto de Salud Carlos III, Majadahonda, 28220 Madrid, Spain; 2Unidades Centrales Científico-Técnicas, Instituto de Salud Carlos III, Majadahonda, 28220 Madrid, Spain; 3Unidad de Microscopía Confocal, Instituto de Salud Carlos III, Majadahonda, 28220 Madrid, Spain; 4Centro de Investigaciones Biológicas Margarita Salas, Consejo Superior de Investigaciones Científicas, 28040 Madrid, Spain; 5Presidencia, Consejo Superior de Investigaciones Científicas, 28006 Madrid, Spain

**Keywords:** DNA supercoiling, DNA topoisomerase I, DNA gyrase, supercoiling regulation, supercoiling homeostasis, nucleoid-associated proteins, topsoisomerase regulator, StaR

## Abstract

The DNA topoisomerases gyrase and topoisomerase I as well as the nucleoid-associated protein HU maintain supercoiling levels in *Streptococcus pneumoniae*, a main human pathogen. Here, we characterized, for the first time, a topoisomerase I regulator protein (StaR). In the presence of sub-inhibitory novobiocin concentrations, which inhibit gyrase activity, higher doubling times were observed in a strain lacking *staR*, and in two strains in which StaR was over-expressed either under the control of the ZnSO_4_-inducible P_Zn_ promoter (strain Δ*staR*P_Zn_*staR*) or of the maltose-inducible P_Mal_ promoter (strain Δ*staR*pLS1ROM*staR)*. These results suggest that StaR has a direct role in novobiocin susceptibility and that the StaR level needs to be maintained within a narrow range. Treatment of Δ*staR*P_Zn_*staR* with inhibitory novobiocin concentrations resulted in a change of the negative DNA supercoiling density (σ) in vivo, which was higher in the absence of StaR (σ = −0.049) than when StaR was overproduced (σ = −0.045). We have located this protein in the nucleoid by using super-resolution confocal microscopy. Through in vitro activity assays, we demonstrated that StaR stimulates TopoI relaxation activity, while it has no effect on gyrase activity. Interaction between TopoI and StaR was detected both in vitro and in vivo by co-immunoprecipitation. No alteration of the transcriptome was associated with StaR amount variation. The results suggest that StaR is a new streptococcal nucleoid-associated protein that activates topoisomerase I activity by direct protein-protein interaction.

## 1. Introduction

The *Streptococcus pneumoniae* chromosome is confined within the nucleoid, in which it is compacted by up to 1000-fold [1], allowing 1.3 Mb to be accommodated within cells just 1–2 μm long. Chromosome topology depends on DNA supercoiling (Sc), which is controlled by DNA topoisomerases [2]. *S. pneumoniae* possesses three of these enzymes: two type II topoisomerases (topoisomerase IV and gyrase), which cleave both DNA strands, and a single type I enzyme (topoisomerase I, TopoI), which cleaves only on one strand. Gyrase and TopoI actively regulate the degree of Sc solving topological problems associated with dynamic DNA remodeling in *Escherichia coli* [3,4,5,6] as well as in *S. pneumoniae* [7,8,9]. In addition, in other bacteria, a number of nucleoid-associated proteins (NAPs) [10] also control Sc, forming a functional network that maintains DNA topology via bending, wrapping, bridging and constraining supercoils [11]. Transcription also regulates Sc, given that domains of negative and positive Sc are generated, behind and ahead of the moving RNA polymerase (RNAP), respectively [12]. In fact, physical interaction between TopoI and RNAP has been detected in vitro in both *E. coli* [13] and *S. pneumoniae* [14]. In addition, ChIP-Seq experiments have revealed in vivo co-localization of RNAP and TopoI in *Mycobacterium tuberculosis* [15] and *S. pneumoniae* [14], as well as co-localization of RNAP and gyrase in *M. tuberculosis*. Likewise, Sc also regulates transcription, structuring the bacterial chromosome into domains with intrinsic topological behavior [7,16,17,18,19]. Genes within the Sc domains of the pneumococcal chromosome have a coordinated transcription [7,16] and similar functions [20]. Four types of domains, defined by their transcriptional response to DNA relaxation and their AT content, have been identified: up-regulated, down-regulated, non-regulated and AT-rich. The genes encoding the topoisomerases are themselves located in regulated Sc domains: *topA* in a down-regulated domain [7], and *gyrB* in an up-regulated domain [21]. The control of Sc in *S. pneumoniae* occurs mainly through the regulation of transcription of the topoisomerase genes. Relaxation triggers up-regulation of gyrase and down-regulation of topoisomerases I and IV, while hyper-negative Sc down-regulates the expression of TopoI. Therefore, TopoI plays a fundamental role in the regulation of transcript levels by Sc, since transcription levels of *topA* in homeostasis correlate with the induced variation in the density of Sc [16].

*S. pneumoniae* is a devastating infectious human pathogen, causing community-acquired pneumonia, meningitis, bacteremia and otitis media and producing the death of one million children worldwide annually [22]. Although resistance in this bacterium to beta-lactams and macrolides antibiotics, which are directed against cell-wall and protein synthesis, respectively, has spread globally [23], no worrisome levels of resistance to antibiotics directed to DNA topology maintenance (fluoroquinolones) [24], which target the type II DNA topoisomerases, have been detected. However, an increase in resistance may occur in tandem with the increased use of fluoroquinolones either due to alterations in their target topoisomerases genes or to the action of active efflux ([25,26,27,28,29,30]. Therefore, knowledge of the molecular basis of Sc control in this important pathogen is essential for finding new antibiotic targets and adequate antibiotic therapies. As mentioned above, in addition to topoisomerases, Sc is also modulated by NAPs. Although several NAPs have been characterized in the Gram-negative bacterium, *E. coli*, very few have been detected in Gram-positive species, including *S. pneumoniae* [31]. HU [32] and SMC [33] are the only NAPs identified in *S. pneumoniae* so far. HU is essential for growth and for the preservation of Sc in this bacterium [32].

The role of many proteins, including NAPs, in bacterial pathogens remains to be described. A previous study identified 44 hypothetical proteins conserved in *S. pneumoniae* and other Gram-positive pathogens, with potential physiological and biomedical value [34]. One of these proteins, a 324-residue-long polypeptide encoded by the *spr0929* locus, was predicted to interact with three protein partners, and was preliminarily classified as a DNA binding protein [34]. Interestingly, the sequence of this protein is highly conserved and present in virtually all *S. pneumoniae* isolates. Notably, genes coding for Spr0929 homologs are found in other Gram-positive pathogens, including *Clostridium*, *Enterococcus* and *Listeria* genera. In this study, we investigated the role of this hypothetical protein in *S. pneumoniae*. We demonstrated that the product of *spr0929* localizes to the nucleoid and activates TopoI relaxation activity affecting Sc homeostasis when DNA relaxation increases. This topoisomerase regulator of *S. pneumoniae* can be considered a new NAP and will be herein renamed as StaR (streptococcal topoisomerase activity regulator).

## 2. Results

### 2.1. The Absence of StaR Affects Novobiocin Susceptibility in S. pneumoniae

To investigate the role of the hypothetical protein encoded by *spr0929*, herein named *staR*, we performed an InterPro analysis to search for bacterial homologues. The best hit found was the NA37 (Nucleotide-Associated proteins of 37 kDa) family. The YejK protein of *E. coli*, the representative member of this family and the only one with experimental information, had 19.4% identity and a 46.6% similarity with pneumococcal StaR (Figure 1A). YejK has been described as a NAP [35]. To decipher the function of StaR, the growth of R6 (wild type) and the previously constructed Δ*staR* deletion mutant [34] was analyzed with or without NOV. NOV inhibits the ATPase activity of gyrase, which is necessary for its Sc activity. However, gyrase has relaxing activity in the absence of ATP hydrolysis, which could also contribute to the final effect in Sc relaxation induced by NOV [36,37]. While in the absence of NOV, both strains showed equivalent growth rates (Figure 1B); in the presence of sub-inhibitory NOV concentrations (0.5 × MIC), Δ*staR* grew slower than R6: doubling time 112 ± 6 min versus 89 ± 6 min (*p* = 0.01), suggesting a role of StaR in Sc maintenance under challenging conditions. In order to purify and characterize StaR, a chromosomal fragment containing its coding gene (*staR*) was cloned into the *E. coli* pET28a BL21-CodonPlus (DE3) expression system, and overproduction of histidine-tagged Star (H_6_-StaR) was achieved by induction with IPTG (Figure 1C). A protein with an apparent molecular weight of 37.1 kDa, corresponding to the predicted molecular weight of the H_6_-StaR protein (39.3 kDa: 37 kDa + 6 His), was detected and further purified by affinity chromatography. Protein from fraction 10 in Figure 1C was collected and used in subsequent in vitro experiments.

### 2.2. The Expression Level of StaR Affects Novobiocin Susceptibility

To elucidate the role of StaR in NOV susceptibility, the Δ*staR*P_Zn_*staR* strain was constructed by introducing an ectopic copy of *staR* under the control of a promoter inducible by zinc (P_Zn_) in the Δ*staR* strain. The growth of this strain was analyzed in the presence of 150 µM ZnSO_4_ as the inductor of *staR* transcription and compared to R6 as control. In addition, the levels of StaR and topoisomerases involved in Sc homeostasis (TopoI and gyrase) were determined by Western blot along the growth curve. Quantification was performed using RpoB as an internal control since no change in its transcription was observed under NOV treatments [7]. While, in the absence of NOV, these strains showed equivalent growth rates both in the absence and in the presence of ZnSO_4_, in the presence of 0.5 × MIC of NOV, Δ*staR*P_Zn_*staR* grew slower than R6 (Figure 2A). The doubling times in the absence of ZnSO_4_ were 215 min ± 32 and 94 min ± 10 for Δ*staR*P_Zn_*staR* and R6 (average ± SD, *n* = 3), respectively. In the presence of ZnSO_4_, these differences were similar: 154 min ± 19 for Δ*staR*P_Zn_*staR* and 89 min ± 10 for R6. Growth in the presence of 150 µM of ZnSO_4_ yielded StaR increases of about 280-fold at all times considered, either in the presence or absence of NOV. However, StaR levels in strain Δ*staR*P_Zn_*staR* with ZnSO_4_ were only about 2-fold higher than those in R6 (Figure 2B,C). Thus, a 2-fold increase in the amount of StaR as well as its absence produced a higher susceptibility to NOV.

Regarding expression levels of topoisomerases, treatment with NOV decreased the level of TopoI about 4-fold at time points 90 and 150 min, and 6-fold at 210 min, in R6 compared to non-treated cells. This was expected considering previous results [7]. Similar decreases were observed in the strain Δ*staR*P_Zn_*staR*, which showed a reduction of about 7-fold in TopoI level in NOV-treated versus non-treated cells at all times tested (90, 150, and 210 min; Figure 2B,C). Therefore, the differences observed in NOV susceptibility between both strains were not associated with a topoisomerase unbalance, but they were due to the distinct levels of StaR as a consequence of its ectopic location. Therefore, appropriate levels of StaR are important for the Sc homeostatic response against the topological stress imposed by sub-inhibitory NOV concentrations [7].

To test whether higher levels of StaR affect cell viability, we used a plasmid cloning approach. We cloned *staR* into pLS1ROM, rendering pLS1ROMstaR, in which *staR* is under the control of the maltose-inducible P_M_ promoter. Accordingly, StaR was induced in the presence of maltose independently on NOV treatment (Figure 3B). The growth rate and protein levels of Δ*staR* strain containing pLS1ROM*staR* were analyzed upon induction with maltose in the presence or in the absence of NOV. Growth in the presence of 0.4% sucrose + 0.4% maltose (SM) yielded a StaR increase of about 10-fold at 150 and 210 min with respect to sucrose (S)-grown cultures, although no effect on NOV susceptibility was observed (Figure 3A). However, induction with 0.8% maltose (M) increased NOV susceptibility, a condition yielding the increase of StaR to about 30-fold (150 min and 210 min) with respect to S-grown cultures (Figure 3B,C). StaR protein levels after induction with SM were similar to those observed in R6, consistent with the absence of any effect on growth (Figure 3B). In contrast, treatment with M induced a 3.5-fold change at all times tested. Altogether, our data indicate that NOV susceptibility increased both in the absence of StaR (Figure 1 and Figure 2A) and when its levels were increased by 2- to 3.5-fold (Figure 2 and Figure 3).

These results suggest that StaR has a direct role in resolving DNA relaxation stress and that its levels need to be precisely regulated. In fact, *staR* is located in a DOWN domain in the R6 genome, and its transcription decreases more than 2-fold under NOV treatment [7]. This explains the increased NOV susceptibility observed when *staR* is ectopically located in Δ*staR*P_Zn_*staR* into a locus (*bgaA*) not located in a supercoiling domain, as well as when it is overproduced in the presence of ZnSO_4_ or maltose.

### 2.3. The Supercoiling Level Is Affected by StaR under Novobiocin Treatment

As part of the Sc homeostasis response, a decrease in TopoI is produced after NOV treatment (Figure 3), coincidentally with a decrease in *topA* transcription [7]. The effect of StaR levels in DNA Sc was analyzed by measuring Sc density of pLS1 in Δ*staR*P_Zn_*staR* with or without NOV. We analyzed the topoisomer distribution of the internal plasmid pLS1 by two-dimensional agarose electrophoresis, a suitable approach for studying Sc levels, which is a reflection of nucleoid compaction [7]. This technique allows separation of DNA molecules by mass and shape as an estimation of the chromosomal Sc. As expected, treatment of Δ*staR*P_Zn_*staR* with NOV resulted in DNA relaxation. In the absence of StaR expression (i.e., no ZnSO_4_ in the medium), the value of negative DNA Sc density (σ) was −0.056, and it decreased to −0.051 and −0.049 at 0.5 × MIC and 1 × MIC NOV, respectively (Figure 4A). However, under StaR overexpression (150 µM ZnSO4) this variation in σ was higher: from −0.057 to −0.048 and −0.045 at 0.5 × MIC and 1 × MIC NOV, respectively (Figure 4B). Therefore, under inhibitory concentrations of NOV, negative Sc density was higher without StaR (σ = −0.049) than with a 2-fold overproduction of StaR (σ = −0.045), *p* = 0.01. These results suggest that StaR increases the level of NOV-induced DNA relaxation.

### 2.4. StaR Is a Nucleoid-Associated Protein

To investigate the possible association of StaR with the nucleoid, we performed Stimulated Emission Depletion (STED) super-resolution microscopy of the wild-type strain (R6). After fixation, the nucleoids were stained with Sytox orange and StaR was immunofluorescently labeled. A total of 66.2% of cells (*n* = 4912) expressed detectable levels of StaR. In these, co-localization was readily observed, indicating that StaR localizes to the nucleoid (Figure 5A). To confirm co-localization, images of these cells were used to calculate the Pearson’s correlation index (Figure 5C). Thresholds corresponding to the average intensity of the green signal of the Δ*staR*P_Zn_*staR* strain (0.042) and a Pearson correlation index of 0.5 were applied. Under these criteria, StaR co-localized with the nucleoid in 51.5% (1674) of the R6 cells analyzed (Figure 5C). As a control, Δ*staR*P_Zn_*staR* cells were imaged under the same growth, fixation and staining conditions in the absence of ZnSO_4_ to image StaR-depleted cells (Figure 5B). Consistently, the distribution of the background green signal detected drastically differed compared to that of R6. The Δ*staR*P_Zn_*staR* strain showed brilliant green areas without any conserved pattern that could be attributed to nonspecific binding of antibodies, and a low-intensity green signal which could correspond to cellular auto-fluorescence (Figure 5B). Moreover, under the same co-localization criteria that was applied to the wild-type (R6) strain, none of the Δ*staR*P_Zn_*staR* cells analyzed showed any co-localization (Figure 5C). This further reinforces the co-localization analysis and shows that, like its homolog YejK, StaR associates with the nucleoid and must be considered a new NAP in *S. pneumoniae*.

### 2.5. StaR Specifically Activates S. pneumoniae TopoI

In vivo results of Sc variation in the presence or absence of StaR suggest a role of this protein in TopoI activation, since DNA relaxation in the presence of NOV is higher when StaR is overproduced. Alternatively, this protein could further inhibit DNA gyrase leading to the same observations.

The effect of StaR on the enzymatic relaxing activity of TopoI was tested. Results showed that StaR activates the relaxation of pBR322 by TopoI in vitro, depending on the amount of StaR used (Figure 6A). We have quantified the activation of TopoI by measuring both decrease of CCC plasmid form and increase of topoisomers (Figure 6B). Since the gel running conditions do not allow for differentiation between OC and RC, both topoisomers with intermediate linking numbers as well as OC/RC forms were considered for calculating TopoI activity. The results showed the simultaneous decrease of CCC and increase of topoisomers as the StaR concentration increases. The CCC and topoisomers reached 2.5% and 97.5%, respectively, at the highest StaR concentration. Moreover, activation of TopoI by StaR depended on the incubation time (Figure 6C). In contrast, StaR did not show topoisomerase activity by itself at concentrations equal to or higher than those used in TopoI activity assays (Figure 6D). In contrast, no activation of *S. pneumoniae* DNA gyrase activity was observed, since 73% of CCC plus topoisomers was observed at all StaR concentrations tested (from 0.25 to 4 µM). Although an excess of StaR over TopoI is required for activation, these results suggest that StaR is a specific activator of TopoI.

### 2.6. Physical Interaction between StaR and TopoI

The specific activation of TopoI by StaR suggested a protein–protein interaction between them. To verify this, in vitro co-immunoprecipitation assays (Co-IP) were carried out with the purified proteins. The two proteins were incubated together and immune-precipitated using polyclonal antibodies against either TopoI or StaR. The presence of TopoI in the anti-StaR pull-down fraction and the presence of StaR in the anti-TopoI pull-down fraction (Figure 7A) indicated a direct interaction between these proteins in vitro.

We also performed Co-IP in vivo after formaldehyde crosslinking (Figure 7B). We used extracts from cultures of Δ*staR*P_Zn_*staR* grown in either the absence or presence of ZnSO_4_, i.e., in the absence or presence of StaR. Western blots showed that StaR-Ab was able to pull down TopoI. Three bands were observed in the Western blot when the StaR-antibody was used in the Co-IP: a band presumably corresponding to TopoI (79 kDa), the band corresponding to StaR (37 kDa) and another band with an apparent molecular weight of 116 kDa, which may correspond to the TopoI-StaR complex. The last band corresponded to 26% of the total amount of the immunoprecipitated StaR. However, no pulldown of StaR by TopoI-Ab was observed, suggesting that the polyclonal antibody against TopoI dissociates the TopoI-StaR complex.

### 2.7. No Global Changes in Transcriptome Are Induced by the Absence or Presence of StaR

The transcriptome of the strain Δ*staR*P_Zn_*staR* was analyzed either in the absence of StaR (no ZnSO_4_ added) or in its presence (150 µM ZnSO_4_). In the presence of the P_Zn_ inductor, 34 differentially expressed genes (DEGs) were detected (Table 1), with the same number of genes (17) up- or down-regulated. As expected, genes known to be regulated by high concentration of ZnSO_4_ [7] were found among these DEGs, with seven up-regulated and six down-regulated. Up-regulated genes included the *psaBCA* operon coding a Mn/Zn transporter, *czcD* coding a cation efflux protein, *prtA* coding a virulence factor, and *adhB* involved in carbon metabolism, while *celA* was down-regulated. Consistent with the increased SatR levels detected in Western blots (280-fold increase, Figure 2) upon ZnSO_4_ induction, mRNA levels of *staR* also increased about 400-fold.

When the effect of NOV on the transcriptome in the absence of StaR (−Zn) was compared with its effect when StaR was overexpressed (+Zn), no important differences were observed (Figure 8A). A total of 324 (110 up-regulated) and 313 (108 up-regulated) DEGs were detected in the absence or presence of StaR, respectively, with 214 DEGs common to both conditions (Figure 8A). However, a slightly differential response was observed, since about one-third of the DEGs under each condition was specific. In the absence of StaR, NOV triggered the expression of 110 genes (67 down-regulated and 43 up-regulated) that did not change in the presence of StaR. Similarly, when StaR was over-expressed, NOV changed the expression of 99 genes (58 down-regulated and 41 up-regulated) that did not change when *staR* was repressed. A comparison of the functional classes of the DEGs did not present significant differences between both conditions. In addition, no difference in the location of DEGs along the genome was detected in the absence or presence of ZnSO_4_ (Figure 8B).

These results suggest that the overexpression of StaR does not affect global transcription induced by NOV in a chromosomal-domain manner, and thus discard StaR as a transcriptional regulator.

## 3. Discussion

The existence of protein networks governing DNA chromosomal topology of pathogens remains an unknown field. Sc levels, nucleoid compaction and viability in *S. pneumoniae* are determined by the balance between gyrase and TopoI activities [7]. This bacterium triggers homeostatic transcriptional responses when Sc density decreases by 25% [8] or when it increases by 40% [16]. These global responses affect transcription of topoisomerases. Relaxation triggers up-regulation (about 2-fold) of gyrase genes (*gyrA* and *gyrB*) and down-regulation of both TopoI (*topA*, around 10-fold) and TopoIV (*parE* and *parC*, around 3-fold) [7]. When Sc increases, *topA* is again down-regulated (about 2-fold), while *gyrA, gyrB, parE* and *parC* remain unchanged [16]. In this way, the down-regulation of *topA* transcription, although to different levels, allowed cell growth and the recovery of Sc.

In addition, the transcriptomic response of wild-type strain R6 to DNA relaxation, which is triggered by NOV, also involved down-regulation (about 2-fold) of the gene coding StaR (*spr0929*), which is indeed included in a down-regulated domain [8]. A similar decrease in transcription (around 3-fold) was observed for the gene coding HU, the only NAP described so far in *S. pneumoniae*, which is essential for Sc maintenance [32]. However, the down-regulation of HU and StaR only occurred at inhibitory NOV concentrations. No change in the transcription or amount of StaR (Figure 2) was observed at sub-inhibitory NOV concentrations. Nevertheless, the location of StaR in a down-regulated domain is important for cell survival in the presence of NOV (Figure 1). This location is consistent with the homeostatic response to relaxation, which involves the down-regulation of *topA* [7], and to the decrease of TopoI levels detected in Western blots (Figure 2). Therefore, relaxation triggers down-regulation of both *topA* and *staR*. This would contribute to the recovery of supercoiling levels after relaxation, since the levels of StaR, which would normally activate TopoI, would decrease. Accordingly, the overproduction of StaR observed in Δ*staR*P_Zn_*staR* upon addition of ZnSO_4_ reduces recovery of Sc at inhibitory (1 × MIC) concentrations of NOV: 21% of Sc variation after 30 min of NOV treatment versus 12.5% without induction (Figure 4). This is presumably due to TopoI activation by StaR. The role of StaR would be the activation of TopoI by protein–protein interaction. Although this activation would hypothetically affect transcription, a global effect was not detected in the in the RNA-Seq experiments carried out in the present paper at the concentration of NOV used.

*S. pneumoniae* lacks most of the NAPs characterized in *E. coli*, with the exception of HU [32] and SMC [33]. Using DAPI to stain DNA and immunostaining for NAPs, it has been shown in *E. coli* that HU and other NAPs localize to the nucleoid [38]. We have recently used super-resolution confocal microscopy and staining of the bacterial nucleoid with the DNA intercalant Sytox [9]. This technique allowed us to accurately identify and locate nucleoids and determine their level of compaction [9]. In this study, we took advantage of this technique to demonstrate that StaR indeed localizes in the nucleoid (Figure 5). A statistical analysis performed with more than 2500 cells showed co-localization of StaR and the nucleoid in more than 50% of analyzed bacteria. Although we cannot compare this number with previous reports, given that, to our knowledge, no statistical quantification has been performed previously for any NAP, we consider that a co-localization higher than 50% indeed reflects a true co-localization. It is important to notice that cultures were not synchronous, and the localization of StaR in the nucleoid could depend on the cell cycle phase. This is the first demonstration of a NAP localizing to the nucleoid in *S. pneumoniae.*

Therefore, StaR should be considered a new NAP, which regulates the activity of DNA topoisomerases. Other NAPs regulating topoisomerases have been identified in other bacteria. This is the case of GyrI, which inhibits *E. coli* gyrase [39], GapR from *Caulobacter crescentus* that stimulates gyrase and Topo IV to remove (+) Sc during DNA replication [40], and HU that activates Topo I in *M. tuberculosis* [41]. *E. coli* YejK, homologous to StaR, has been shown to inhibit both gyrase and TopoIV activities [42]. We have not found gyrase activity alteration by StaR, which probably reflects the low sequence identity between YejK and StaR (Figure 1). Activation of Topo IV has not been tested, given that *S. pneumoniae* Sc is mostly controlled by the opposing activities of Topo I and gyrase [7,16]. Regulation of the topoisomerase type I activity has also been observed in eukaryotes. It has been described that TOP1 activity is controlled by the phosphorylated RNAPII to modulate elongation during transcription [43]. In *S. pneumoniae*, TopoI is the main topoisomerase involved in the regulation of supercoiling [7,16] and in transcription [14]; therefore, its activity must be finely regulated.

While co-membership to the same Pfam family does not imply sharing exactly the same properties, we have confirmed that StaR is a NAP and has a similar size to *E. coli* YejK, indicating that NA37 is a proper global name for this protein family. In future studies, the structure–function relationship may add support at residue-level to the observations described here. Unfortunately, no homologous structural templates for modeling are currently available. Numerous attempts to resolve the StaR structure from *S. pneumoniae* and *Enterococcus faecalis* were unsuccessful since crystals showed low-resolution diffraction. In all, evidence indicate NA37 proteins are not prone to experimental structural analyses. In contrast, novel ab initio modeling tools based on artificial intelligence such as AlphaFold achieve unprecedented accuracy [44]. Such homology-free models may help to evaluate critical traits of StaR. These include the co-location of basic residues able to interact with DNA, the domain delineation, and the identification of interacting sides for docking studies.

To summarize, StaR is a new NAP of *S. pneumoniae* and is the first characterized TopoI modulator in this bacterium, playing an important role in the genome-relaxation-triggered homeostasis. Therefore, we propose that StaR is involved in the fine-tuning control of the activity of the essential TopoI, contributing to the homeostatic response to Sc changes.

## 4. Materials and Methods

### 4.1. Bacterial Strains, Growth Conditions and Transformation

*S. pneumoniae* strains were grown as static cultures at 37 °C in a casein hydrolysate-based liquid medium (AGCH) containing 0.2% yeast extract and 0.3% sucrose [45] (C + Y). *E. coli* was grown in LB medium at 37 °C with agitation. A strain with a deletion in *staR* was constructed as described [34]. Transformation of pneumococcal strains was performed as previously reported [45] and selection of transformants was in 1 µg/mL tetracycline, 250 µg/mL kanamycin, 2.5 µg/mL chloramphenicol, or 1 µg/mL erythromycin. Growth was followed by measuring the OD at 620 _nm_ either with a UV-visible spectrophotometer (Evolution 201, Thermo Scientific, Waltham, MA, USA) or in a microplate reader (Infinite F200, Tecan, Männedorf, Switzerland). Both measures correlate linearly by means of the equation y = 0.2163x + 0.1151 (y = microplate reader measure, x = spectrophotometer), R^2^ = 0.98 [32].

### 4.2. Construction of Strains and Plasmids

To overproduce and purify StaR, a PCR fragment containing its coding gene was amplified from R6 genomic DNA with primers *spr0929*_F (5′-GCGCGCTAGCATGGATATTTATATTAAGAAAGCC) and *spr0929*_R (5′-GCGCGGATCCTTATTTACTTTGGATATCCTCG) [34]. Restriction enzyme sites are showed underlined in all oligonucleotides. The amplicon was digested with NheI and BamHI and cloned into pET28a cut with the same enzymes. The plasmid was transferred to BL21-CodonPlus (DE3) (Agilent Technologies, Santa Clara, CA, USA) *E. coli* cells. To test the physiological effects of the overexpression of *staR*, a genetic fusion between the P_Zn_–ribosome binding site and the *staR* coding region was made to render the *ΔstaRP_Zn_staR* strain. The *staR* gene was amplified with spr0929_ScIF (5′-: GCGCGAGCTCATGGATATTTATATTAAGAAAGCC) and spr0929_SlIR (5′-GCGCGTCGACTTATTTACTTTGGATATCCTCG) oligonucleotides using R6 chromosomal DNA as template. The amplicon was digested with SacI and SalI enzymes, targets included in the oligonucleotides, and introduced into pZK plasmid cleaved with the same enzymes to render plasmid pZK_0929. The pZK plasmid contains the upstream region of the *czcD* gene that shows up-regulation in the presence of zinc [46]. A PCR fragment containing the Zn-inducible promoter upstream the spr0929 ORF was amplified from pZK_0929 with KmR_B1_Rv (5′-CGCGGGATCCAGGATCCATCGATACAAATTCC) and pZK_Xb1_Rv (5′-GCGCTCTAGACACCATAAAAAATGAACTTGG) oligonucleotides. This construction was introduced into the *spr1806* locus of the *ΔstaR* strain. The *spr1806* gene codes for a surface protein, which in laboratory conditions, is not expressed; subsequently, the deletion mutant shows no relevant differences with respect to the wild-type strain [47]. For that, DNA fragments of ~1 Kb corresponding to the exact flanking regions of *spr1806* were amplified with oligonucleotides spr1806_ExF (5′-CATAGGTCTCCACATCGAAA) and spr1806_LiURB1 (5′-GCGCGGATCCAACAGCTAGAAAATTCTATTCT), for the upstream region, and oligonucleotides spr1806_LiDFXb (5′-GCGCTCTAGATACTGAACCATCTGGGAATG) and spr1806_ExR (5′-TTGCCCGTCAAAATGATTTG) for the downstream region. Amplicons were cut with BamHI and XbaI, respectively, and simultaneously ligated to the fragment containing the Zn-inducible promoter and spr0929 ORF. The ligation fragment was introduced into pre-competent Δ*staR* cells by natural transformation and transformants were selected with 50 µg/mL kanamycin. Recombination was checked by PCR.

To test the role of *staR* overproduction, the gene was cloned into pLS1ROM vector under the control of P_Mal_ promoter, which is controlled by MalR transcriptional repressor that is cloned in pLS1ROM under the control of the constitutive P_tet_ promoter [48]. For this purpose, a PCR fragment containing *staR* was obtained from R6 chromosome using oligonucleotides 0929HindF (5′-CGCAAGCTTGGAGGACTTTTATGGATATTTATATTAAGAAAGC) and 0929BamR (5′-GCGGATCCGCAAAAAGGAAGACTGCTAGTACAAG), cut with HindIII and BamHI, and ligated to pLS1ROM cut with the same enzymes. The constructed plasmid pLS1ROMstaR was transformed into Δ*staR* strain and transformants were selected with 1 µg/mL erythromycin.

### 4.3. Cloning, Expression and Purification of Proteins

H_6_-StaR was purified by His-affinity chromatography using HiTrap Chelating HP columns (Amersham-Merck, Madrid, Spain) and an AKTA prime system (Amersham Bioscience, Amersham, UK) from *E. coli* BL21-CodonPlus (DE3) cells harboring pET28-StaR. The strain was grown in LB medium containing 50 µg/mL kanamycin at 37 °C to OD_600 nm_ = 0.6 and *staR* transcription was induced with 1 mM isopropyl thio-β-D-galactoside (IPTG) for 1 h. The cells were harvested by centrifugation, washed with phosphate-buffered saline (PBS) and suspended in 72 mL of buffer A (50 mM Tris-HCl pH 8.0, 0.5 M NaCl). The cells were sonicated (8 cycles of 20 sec ON/60 sec OFF) and cell extracts were clarified by centrifugation at 100,000× *g* for 1 h at 4 °C. A two-step protein fractionation with ammonium sulfate at 40% pellet and 60% saturation was performed. The pellet from 60% saturation was suspended in 51 mL of buffer A containing 40 mM imidazole and applied to a 5 mL HiTrap column equilibrated buffer A with 40 mM imidazole. Protein elution was achieved with 100 mL of a 100-to-500-mM imidazol gradient in buffer A, and fractions were analyzed by SDS-PAGE. Imidazol was removed from samples by dialysis against 20 mM Tris-HCl pH 7.5, 1 mM EDTA, 0.1 M NaCl, 50% glycerol. Protein concentration was determined with a Qubit 4 fluorimeter using the Qubit^®^ Protein Assay Kit (Invitrogen-ThermoFisher Scientific, Waltham, MA, USA). TopoI, GyrA and GyrB were purified by affinity chromatography in a Ni-NTA (Quiagen, Hilden, Germany) column following manufacturer’s instructions as described previously [26].

### 4.4. Confocal Microscopy

The cells were grown until OD_620 nm_ = 0.4, centrifuged and suspended in PBS to a number of 8 × 10^9^ cells/mL. The cells were fixed with 2% formaldehyde for at least 16 h. For immunofluorescence microscopy, 8 × 10^7^ cells were incubated with 5 µM SytoxTM Orange Nucleic Acid Stain (Invitrogen) at room temperature for 5 min and extended into a poly-L-lysine coated glass slide. The cells were permeabilized by immersing the slide in methanol at −20 °C for 10 min. The slide was first incubated with 2% BSA, 0.2% Triton X100 in PBS (BSA-PBST), incubated with a rabbit StaR polyclonal antibody at 1/100 dilution in BSA-PBST buffer, washed with PBS and incubated with the anti-rabbit Abberior^®^ STAR 488 antibody (Abberior, Göttingen, Germany) at 1/200 dilution. After a wash with PBS, the slides were mounted with ProLongTM Gold Antifade Mountant (Invitrogen) and sealed. Nucleoids and StaR were imaged as previously described [9] in a confocal microscope STELLARIS 8–FALCON/STED (Leica Microsystems, Wetzlar, Germany) with a HC PL APO 100×/1.40 NA × OIL immersion objective. Super resolution images were acquired by Stimulated Emission Depletion (STED) microscopy using 660 nm depletion laser. Analysis was performed with Cell profiler software v4.2.5, and co-localization-correlation index was calculated for each bacteria with a custom-made routine.

### 4.5. Western Blot Assays and Antibody Purification

Whole cell lysates (~5 × 10^5^ cells) were obtained by centrifugation of 10 mL cultures (OD_620 nm_ = 0.4). They were suspended in 400 µL of sample loading buffer (0.3 M Tris-HCl pH 6.8, 10% SDS, 50% *v/v* glycerol and 0.05% bromophenol) supplemented with 0.5 M β-mercaptoethanol and incubated for 5 min at 100 °C. Lysates were separated on Any kD™ Criterion™ TGX Stain-Free™ Protein Gels (Bio-Rad, Hercules, CA, USA). They were transferred to 0.2 µm PVDF membranes with a Trans-Blot Turbo Transfer System (Bio-Rad) at 25 V, 1 A for 30 min. Membranes were blocked with 5% skim milk in Tris-buffered saline for 2 h and incubated with anti-TopoI and anti-GyrA [16], anti-RpoB [14] and anti-StaR (diluted 1:1000). Rabbit polyclonal antibody against StaR was obtained from Davids Biotechnologie from 0.5 mg of protein extracted from SDS-gel following a 28-day SuperFast immunization protocol. Anti-rabbit IgG-peroxidase Ab (Sigma-Aldrich-Merck, Madrid, Spain) was used as the secondary Ab. SuperSignal West Pico chemiluminescent substrate (Thermo-Fisher, Waltham, MA, USA) was used to develop the membranes. Signal was detected with a ChemiDocTM MP system (Bio-Rad). Images were analyzed using Image LabTM software (Bio-Rad). Molecular masses of GyrA and TopoI are 92 kDa and 79 kDa, respectively.

### 4.6. DNA Topoisomerase Assays

Relaxation reactions of pBR322 by TopoI were carried out exactly as described previously [49]. Reactions of 200 µL contained 0.5 µg of CCC pBR322 in 20 mM Tris-HCl pH 8, 100 mM KCl, 10 mM MgCl_2_, 1 mM DTT, 50 µg BSA/mL and TopoI at the indicated concentrations. Incubation with TopoI was at 37 °C for 1 h and the reaction was terminated by 2 min incubation at 37 °C with 50 mM EDTA. Then an additional incubation of 1 h at 37 °C with 1% SDS, 100 µg/mL proteinase K was performed. Reaction products were ethanol precipitated, suspended in electrophoresis loading buffer and analyzed by electrophoresis in 1% agarose gels run at 18 V for 18 h. DNA quantification of agarose gels was done by scanning densitometry after electrophoresis and ethidium bromide staining. Quantification of TopoI activity was calculated by gel densitometry using the Image Lab program (Bio-Rad Laboratories, Hercules, CA, USA). To calculate activity, the OC form amount was determined and divided by the total amount of DNA in each well. Enzymatic reactions with gyrase were conducted as described previously [50].

### 4.7. Co-Immunoprecipitation (Co-IP) Assays

For in vitro Co-IP, the buffer used along the experiment was Co-IP buffer (10 mM sodium phosphate pH 7.4, 50 mM NaCl, 0.5% Nonidet P40). Purified TopoI (60 ng/µL) was dialyzed against the buffer, and StaR (9.2 µg/mL) was diluted 1000-fold in the same buffer. A total of 2 µg of TopoI and StaR, respectively, was mixed into 200 µL of buffer for 2 h at 4 °C with rotation at room temperature. Purified Abs (4 µg) were bound to Dynabeads^®^ Protein G for 2 h in the buffer at room temperature with rotation as described by the manufacturer and were added to the protein mix. After incubation, the mix was washed 3 times and it was collected from the magnetic grid with SDS-loading buffer without β-mercaptoethanol. The samples were boiled for 5 min, loaded in a 4–20% SDS-polyacrylamide gel and run for 1 h at 170 v. Gels were stained with Coomassie blue for about 1 h.

For in vivo Co-IP, 10 mL of fixing buffer (50 mM Tris-HCl pH 8.0, 100 mM NaCl, 0.5 mM EGTA, 1 mM EDTA, 11% (*v*/*v*) formaldehyde) was added to 100 mL of culture grown to OD_620 nm_ = 0.4 (7 × 10^9^ cells) with or without 150 µM ZnSO_4_. This mix was incubated for 30 min at room temperature. Crosslinking was stopped by adding 10 mL of cold quenching solution (1.25 M glycine in 50 mM Tris-HCl pH 8.0, 100 mM NaCl, 0.5 mM EGTA, 1 mM EDTA). The mix was transferred to an ice/water bath and rotated for 30 min at 4 °C. Cells were collected by centrifugation at 4 °C and suspended in 50 mL ice-cold PBS. Washing was repeated twice. Finally, cells were collected by centrifugation and suspended in 1 mL of lysis buffer (50 mM Hepes-KOH pH 7.5, 140 mM NaCl, 1 mM EDTA, 1% (*v*/*v*) Triton X-100, 0.1% (*w*/*v*) sodium deoxicholate, 1 mM PMSF), which contained 100 µg/mL of RNase A. Cells were sonicated using 25 cycles (30 s on/30 s off) in a Bioruptor^®^ Pico sonicator (Diagenode, Ougrée, Belgium). The sonicated suspension was centrifuged at 21,000× *g* for 10 min at 4 °C. 100 µL of the supernatant was kept at −20 °C as whole cell extract control.

TopoI and StaR antibodies were purified as described previously [14]. The antibodies (10 µg) were bound to 50 µL Dynabeads^®^ Protein G. For immunoprecipitation, 500 µL of sonicated suspension (antigens) was added to 200 µL of the magnetic-bead–Ab complex and incubated on a rotating mixer for 4 h at 4 °C. The tube was placed on a magnet and the supernatant was removed. It was washed once with the lysis buffer, once with the lysis buffer containing 500 mM NaCl and once with the lysis buffer containing 250 mM LiCl. The supernatant was removed and proteins bound to Dynabeads were resupended in 30 µL of the sample loading buffer containing 50 mM glycine. Dynabeads^®^ Protein G—antigen was incubated at 65 °C overnight with shaking to elute DNA from the beads and reverse cross-links. Samples were fractionated in SDS/PAGE and proteins were detected by Western blot using TopoI-Ab (1:500 dilution) and StaR-Ab (1:1000 dilution) antibodies.

### 4.8. Analysis of the Topology of Plasmids

Plasmid DNA topoisomers were analyzed in neutral/neutral two-dimensional agarose gels. The first dimension was run at 1.5 V/cm in a 0.4% agarose (Seakem-Lonza, Basel, Switzerland) gel in 1 × Tris-borate-EDTA (TBE) buffer for 20 h at room temperature. The second dimension was run at 7.5 V/cm in 1% agarose gel in 1 × TBE buffer for 7–9 h at 4 °C. Chloroquine (Sigma-Aldrich-Merck, Madrid, Spain) was added to the TBE buffer in both the agarose and the running buffer. Chloroquine is a DNA intercalating agent that removes negative Sc in bacterial plasmids. Increasing the concentration of chloroquine progressively eliminates negative Sc until the plasmid is relaxed and can then introduce net positive Sc. In this way, the use of adequate concentrations of chloroquine during each dimension in the 2D analysis allows the efficient resolution of the different topoisomers [51]. Chloroquine concentrations used were 1 µg/mL and 2 µg/mL in the first and second dimension, respectively. Gels were stained with 0.5 µg/mL ethidium bromide for 1 h at room temperature. Images were captured in a ChemiDoc Imaging System (Bio-Rad) and analyzed with the Image Lab software (BioRad). DNA supercoiling density (σ) was calculated from the equation σ = ΔLk/Lk_0_. Linking number differences (ΔLk) were determined using the equation ΔLk = Lk − Lk_0_, in which Lk_0_ = N/10.5, being N the size of the molecule in bp (4408) and 10.5 the number of bp per one complete turn in B-DNA. To simplify it, σ = Lk of the most abundant topoisomer/(N/10.5). The Lk of the most abundant topoisomer was calculated taking into account that the topoisomer that migrated with Δ*Lk* = 0 in the second dimension has a Δ*Wr* = −14 during the first dimension (the number of positive supercoils introduced by 2 µg/mL chloroquine).

### 4.9. RNA Library Preparation for RNA-Seq

Total RNA (5 µg) obtained from samples containing 2 to 4 × 10^8^ cells with the RNeasy mini kit (Quiagen) was depleted of ribosomal RNA using the Ribo-Zero Magnetic Kit (Bacteria) (Illumina, San Diego, CA, USA). Libraries for RNA sequencing (RNA-Seq) were prepared using the ScriptSeq v2 RNA-Seq system (Illumina). Briefly, 1 µg of RNA samples were chemically fragmented. Using randomly primed cDNA synthesis, cDNAs that were tagged at the 5′ end (equivalent to the 3′ end of the original RNA) were synthesized. The cDNAs were then tagged at the 3′ end using Terminal-Tagging Oligos (TTO). These oligonucleotides randomly annealed to the cDNA and were extended by DNA polymerase. The resulting di-tagged cDNAs were purified with the Ampure bead XP system. The enrichment and barcoding of the purified di-tagged cDNAs were done with 15 cycles of PCR. The library size was determined with a 2100 Bioanalyzer Instrument (Agilent Technologies). qPCR quantifications were done with a Kapa Library quantification Kit (Kapa Biosystems, Sigma-Aldrich-Merck, Madrid, Spain).

### 4.10. RNA Seq Data Analysis

Analysis of RNA-Seq data was carried out using the web-based platform Galaxy [14]. The quality of raw sequence data was analyzed with the FASQC tool. Sequencing reads were mapped against the *S. pneumoniae* R6 genome (ASM704v1) using the BWA software package (Galaxy version 0.7.17.4) in simple Illumina mode. The number of reads overlapping each coding gene was obtained using program feature Count. Count tables were used as input in DESeq2 for the analysis of differential expression. A threshold *p*-value-adjusted of 0.01 was considered.

## Figures and Tables

**Figure 1 ijms-24-05973-f001:**
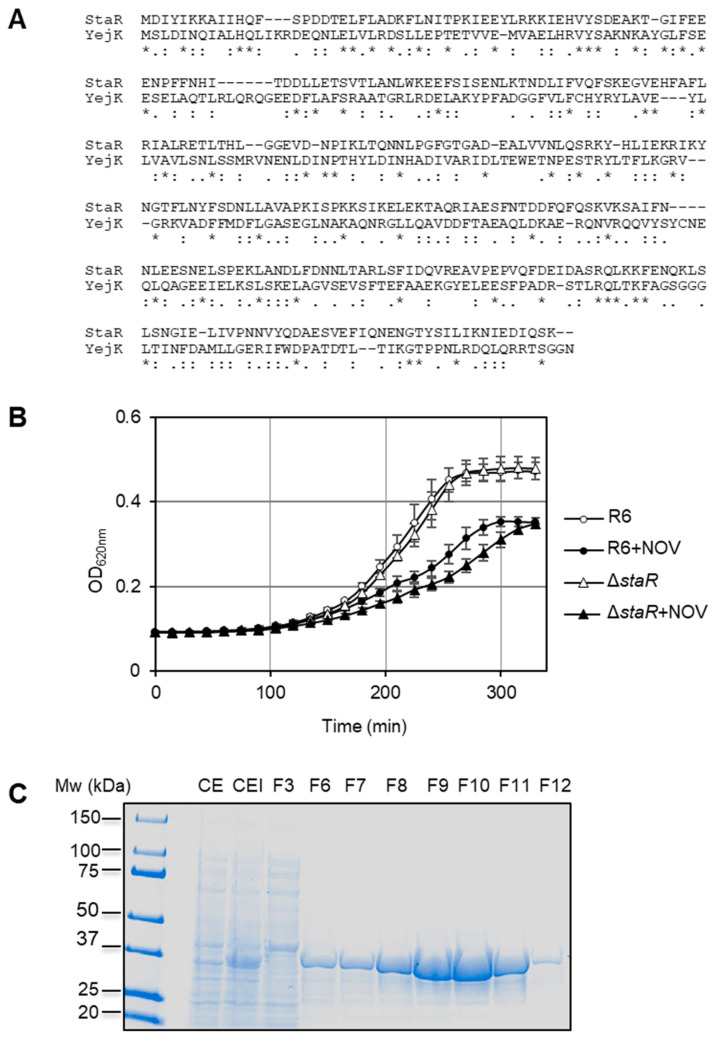
Identification of StaR. (**A**) Alignment between *E.coli* YejK and *S. pneumoniae* StaR using multiple sequence alignment. Asterisks label identical residues between both proteins and colons denote similar residues. (**B**) Absence of *staR* affects NOV susceptibility. Isogenic strains carrying wild-type *staR* (R6) or a deletion of *staR* (Δ*staR*) were grown to OD_620 nm_ = 0.3, diluted 200-fold in the absence or presence of NOV at 0.5 × MIC and their growth was recorded in a TECAN Infinite 200 PRO reader. Data are the average of three independent replicates ± SEM. (**C**) Purification of H_6_-StaR from *E. coli* BL21-CodonPlus (DE3) cells harboring pET28-*staR*. SDS-PAGE of 30 µL of crude extract samples of no-induced (CE) or IPTG-induced (CEI) cultures. Fraction numbers eluted from the AKTA prime system column are indicated; 30 µL of fractions were run in the gel that was stained with Coomassie-blue. Molecular weights are indicated on the left.

**Figure 2 ijms-24-05973-f002:**
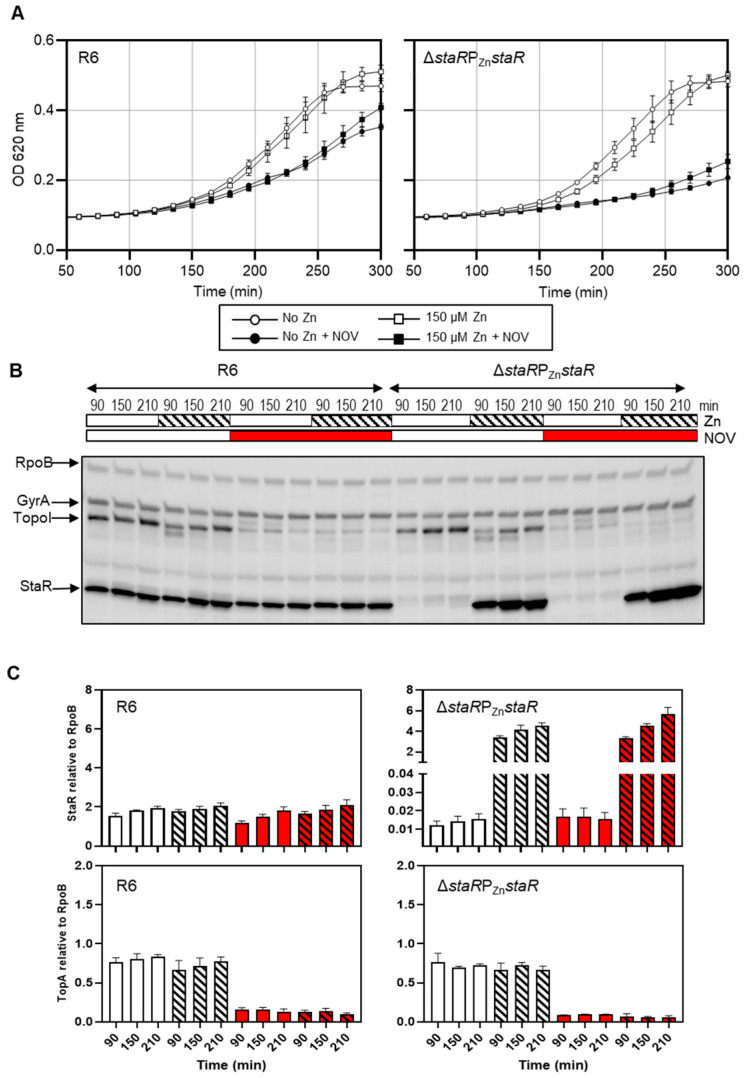
Effect of StaR expression under the control of P_Zn_ promoter. Wild-type R6 and Δ*staR*P_Zn_*staR* strains were used. Strain Δ*staR*P_Zn_*staR* carries a deletion of *staR* and a single ectopic copy of *staR* under the control of P_Zn_ promoter. (**A**) Expression of StaR under the control of P_Zn_ impairs growth under NOV treatment. Strains were grown to OD_620 nm_ = 0.3 in medium without ZnSO_4_, diluted 200-fold in medium without (empty symbols) or with (black symbols) NOV (0.5 × MIC) together with 150 µM ZnSO_4_ (squares) or without ZnSO_4_ (circles) and cultures were grown in a TECAN Infinite 200 PRO reader. (**B**) Western blot assays of one of the three replicates using antibodies against GyrA (1:2000), TopoI (1:500), StaR (1:1000) and RpoB (1:2000) as internal control. Samples from the cultures described in (**A**) were taken at the indicated time points and processed as described in the Materials and Methods section. (**C**) Levels of StaR and TopoI proteins in the absence (empty bars) or in the presence (striped bars) of ZnSO_4_, and in the absence (white bars) or in the presence (red bars) of NOV, as quantified from Western blots. Protein levels relative to RpoB are shown. Data presented are the average of three independent replicates ± SEM.

**Figure 3 ijms-24-05973-f003:**
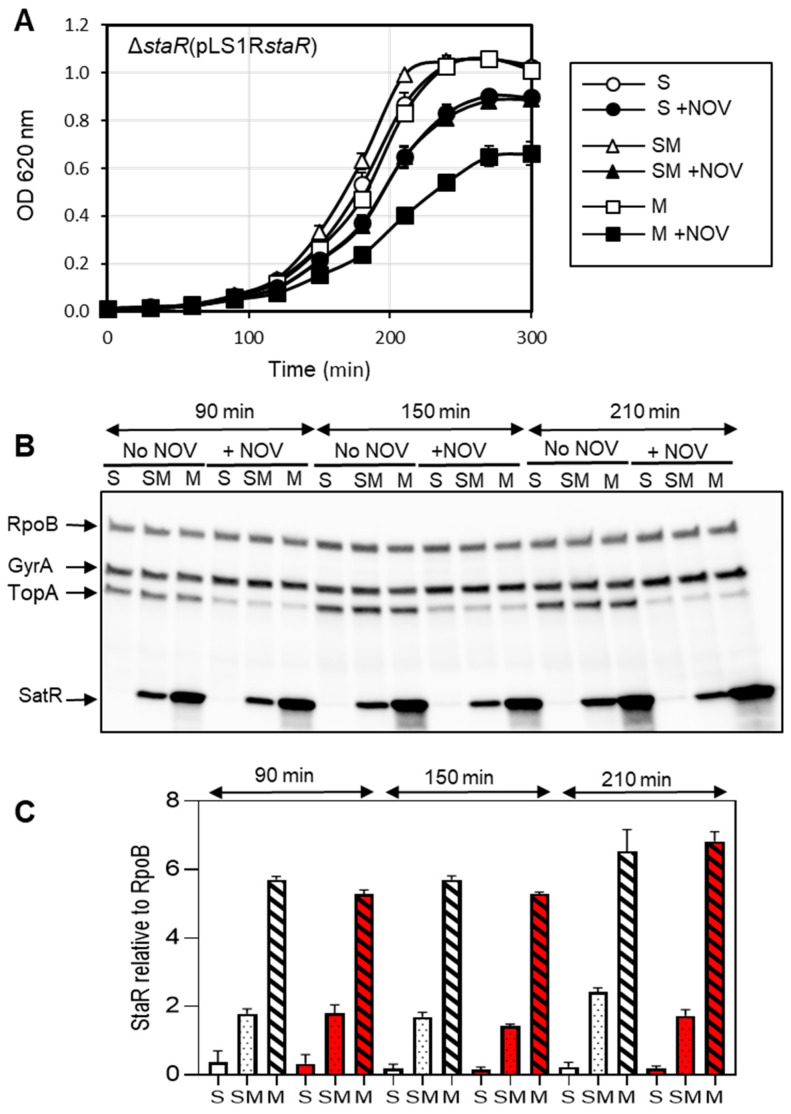
Effect on NOV susceptibility of increasing amounts of StaR. The strain used was Δ*staR* transformed with plasmid pLS1R- *staR*, which carries *staR* under the control of the maltose-regulated promoter P_M_. (**A**) Growth curves in the presence of 0.8% sucrose (S), 0.4% sucrose + 0.4% maltose (SM), or 0.8% maltose (M) and in the absence or presence of NOV at 0.5 × MIC. The bacterial culture was grown to OD_620nm_ = 0.3 in medium with S. Cells were washed and diluted 100-fold in media containing either S, SM or M, in the absence or presence of NOV. (**B**) Western blot assays of GyrA, TopoI and StaR using antibodies against GyrA (1:2000), TopoI (1:500), StaR (1:1000) and RpoB (1:500) as internal control. Samples were collected at the indicated time points and treated as described in the Material and Methods section. A representative gel of three replicates is shown. (**C**) StaR levels in cultures grown with S (empty bars), SM (pointed bars), or M (striped bars), and in the absence (white bars) or in the presence (red bars) of NOV, quantified from Western blots. Protein levels relative to RpoB are shown. Data presented are the average of three independent replicates ± SEM.

**Figure 4 ijms-24-05973-f004:**
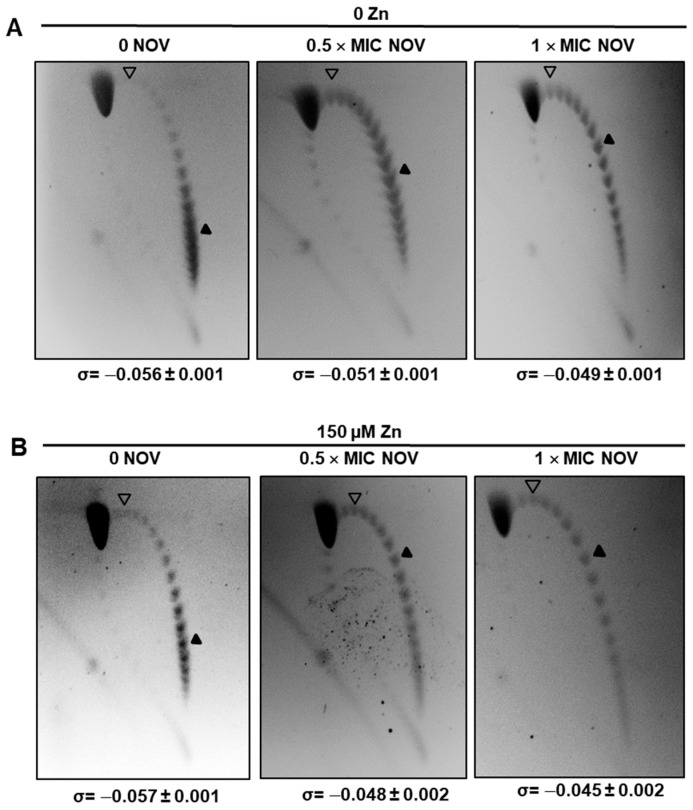
pLS1 topoisomers distribution in strain Δ*staR*P_Zn_*staR* at the indicated ZnSO_4_ and NOV concentrations. Cultures were grown to OD_620 nm_ = 0.2, diluted 50-fold in medium without ZnSO_4_ (**A**) or with ZnSO_4_ (**B**) and grown to OD_620 nm_ = 0.4. The samples taken at this point corresponded to those not treated with NOV. For NOV-treated samples, NOV was added to the cultures for 30 min so that the final concentration of NOV was 0.5 × MIC or 1 × MIC NOV and the ZnSO_4_ concentration remained constant. Plasmids were extracted and run in 2D-agarose gels in the presence of 1 and 2 µg/mL chloroquine in the first and second dimensions, respectively. Values are the mean of four independent replicates ± SD. Gels of a typical experiment are shown. Empty arrowheads indicate the topoisomer that migrated with Δ*Lk* = 0 in the second dimension and has a Δ*Wr* = −14 during the first dimension (the number of positive supercoils introduced by 2 µg/mL chloroquine). Black arrowheads indicate the most abundant topoisomer.

**Figure 5 ijms-24-05973-f005:**
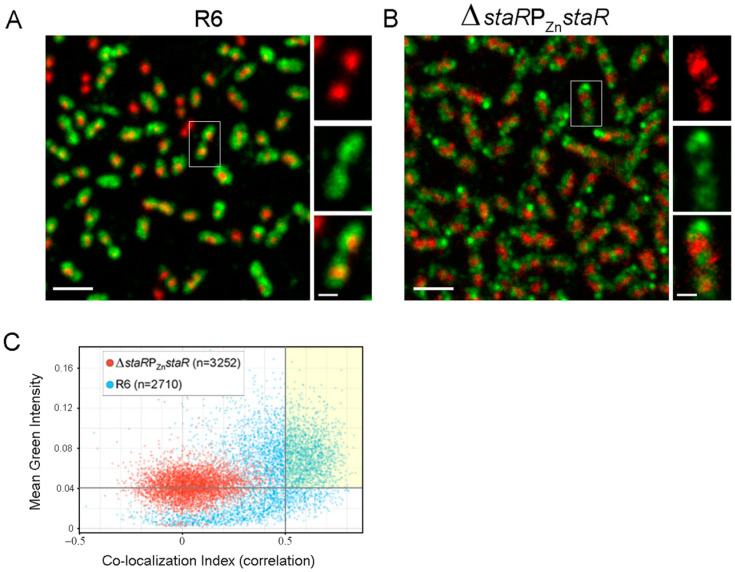
StaR localizes to the nucleoid. Strain R6 and Δ*staR*P_Zn_*staR* were grown in the absence of ZnSO_4_ to mid-log phase (OD_620 nm_ = 0.4). Cells were fixed, the nucleoids were stained with Sytox orange and StaR was immunostained using the Abberior Star 488 anti-rabbit secondary antibody as described in the Materials and Methods section. STED microscopy was used to determine the intracellular localization of nucleoids and StaR. (**A**) General view (left panel) and inset of a representative co-localizing cell of R6, showing nucleoids (red) and StaR labeling (green). Scale bar indicates 2 µm in the general views or 0.5 µM in the insets. (**B**) General view (left panel) and inset of a representative co-localizing cell of Δ*staR*P_Zn_*staR*, as a control of StaR depletion. (**C**) Pearson correlation analysis of the strains indicated in (**A**,**B**). *y*-axis represents the mean intensity of green label (StaR) for Δ*staR*PZn*staR* (red points) and R6 (blue points) and the *x*-axis indicated the Pearson correlation between green and red (nucleoids) labels. Gray lines in the *x*- and *y*-axis correspond to the thresholds applied to calculate the number of cells where StaR co-localizes with the nucleoid. A yellow square highlights the R6 population showing StaR–nucleoid co-localization.

**Figure 6 ijms-24-05973-f006:**
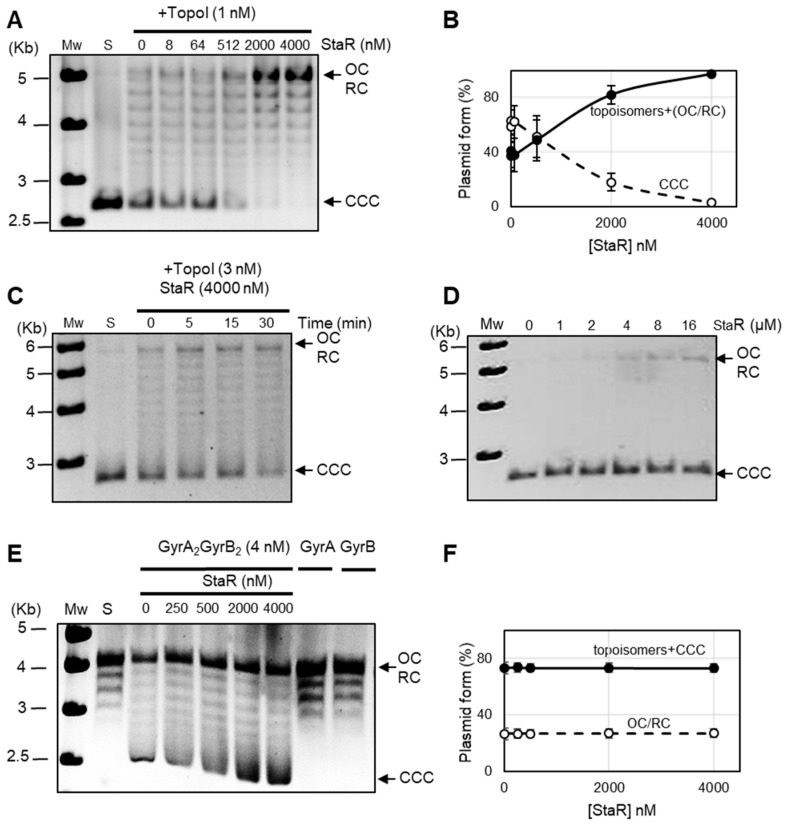
StaR activates the relaxation activity of *S. pneumoniae* TopoI but does not have any effect on gyrase. (**A**) Activation of TopoI as a function of StaR concentration. Plasmid pBR322 was incubated at 37 °C with purified TopoI for 1 h. The indicated StaR concentrations were added simultaneously to the reaction mix. Samples were processed and analyzed as described in the Materials and Methods section. Mw, molecular weight standard (Kb); S, substrate; CCC, covalently closed circles; OC, relaxed open circles; RC, relaxed circular plasmid forms. (**B**) Activity of TopoI determined as increase of the topoisomers (including OC/RC forms) and decrease of CCC form in the presence of StaR. Results are the average ± SEM of three independent replicates. (**C**) Activation as a function of reaction time. Reactions were carried out as in (**A**). (**D**) StaR does not have topoisomerase activity by itself. Reactions were carried out as in A, except that TopoI was not added. Concentrations of StaR are indicated. (**E**) DNA gyrase activity is not affected by StaR. Supercoiling activity over relaxed pBR322 using 4 nM of reconstituted gyrase was assayed in the presence of different StaR concentrations at 37 °C for 1 h. Incubation with GyrA or GyrB alone is included as a control. (**F**) Activity of gyrase determined as percentage of topoisomers +CCC and percentage of OC/RC forms in the presence of StaR. Results are the average ± SEM of three independent replicates.

**Figure 7 ijms-24-05973-f007:**
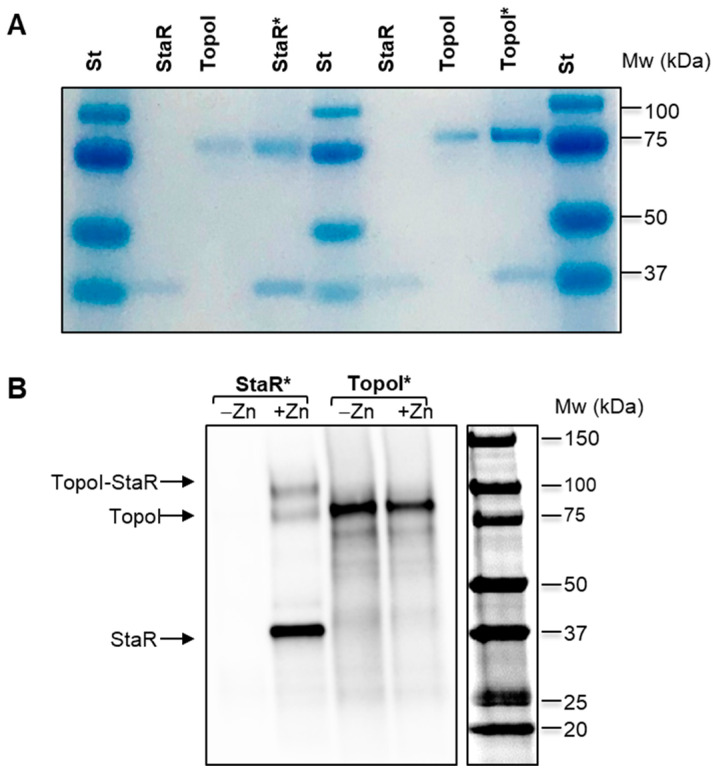
Co-immunoprecipitation of StaR and TopoI. (**A**) In vitro assays. Purified proteins were incubated together in the presence of the antibody against either StaR (StaR*) or TopoI (TopoI*). Samples were treated as described in the Materials and Methods section and run in a 4–20% polyacrylamide gel, which was stained with Coomassie blue. Purified proteins were loaded as controls. (**B**) In vivo assays. Cultures of Δ*staR*P_Zn_*staR* were grown either in the absence (−Zn) or presence (+Zn) of 150 µM ZnSO_4_ to OD_620 nm_ = 0.4, fixed with formaldehyde and treated as described in the Material and Methods section. The samples were run in a 4–20% polyacrylamide gel and Western blots were performed using antibodies against TopoI (1:500 dilution) and StaR (1:1000 dilution). The expected positions of TopoI and StaR proteins and of the TopoI-StaR complex are indicated.

**Figure 8 ijms-24-05973-f008:**
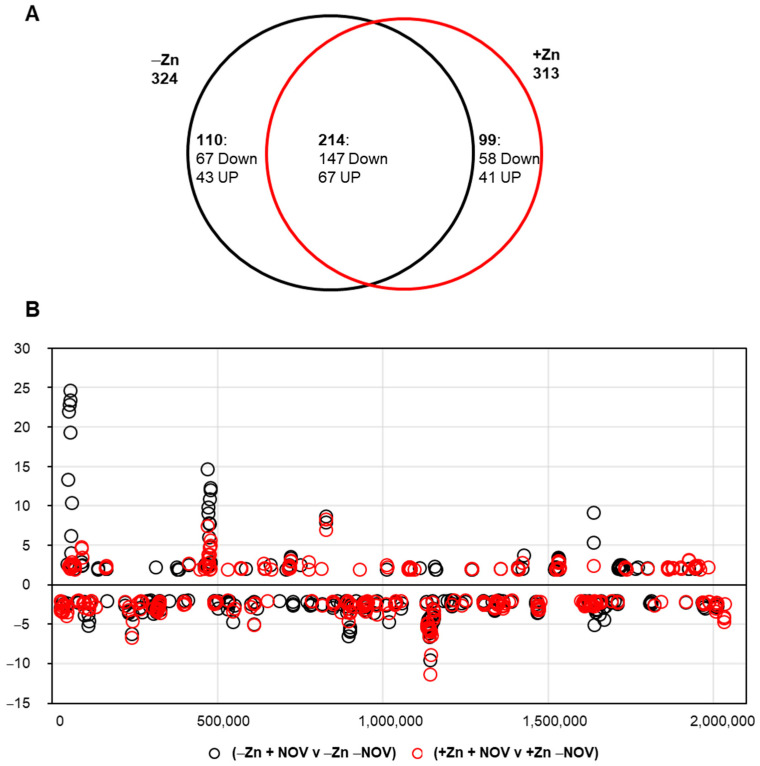
Number and localization of DEGs in the Δ*staR*P_Zn_*staR* chromosome upon treatment with NOV in the presence or absence of StaR. Cultures were grown to OD_620 nm_ = 0.2, diluted 50-fold in medium either without or with 150 µM ZnSO_4_ and grown to OD_620 nm_ = 0.4. Samples taken at this point were considered as non-treated with NOV. NOV-treated samples were grown for another 30 min with 0.5 × MIC of NOV. Total RNA was isolated and sequenced as described in the Materials and Methods section. (**A**) Venn diagram showing DEGs upon treatment with NOV in the absence or presence of ZnSO_4_. (**B**) The relative fold variation of each gene represented against the 5′ location of each open reading frame in the *S. pneumoniae* R6 chromosome (bases 1 to 2,038,615). A fold change ≥ 2 (absolute value) and a *p*-value-adjusted ≤0.01 were considered.

**Table 1 ijms-24-05973-t001:** DEGs detected in Δ*staR*P_Zn_*staR* in the presence of ZnSO_4_ versus in its absence. Cultures were grown to OD_620 nm_ = 0.2, diluted 50-fold in medium either without ZnSO_4_ or with 150 µM ZnSO_4_ and grown to OD_620 nm_ = 0.4.

Role o Subrole	Common Name	R6 Locus (Gene) ^a^	Relative Fold Variation ^b^
Carbohydrate metabolism	Beta-galactosidase	spr0059 (*bgaC)*	*−2.2*
Hypothetical proteins		spr0066	**3.9**
Transport: Carbohydrates	ABCT membrane permease	spr0082	*−2.2*
Hypothetical proteins		spr0085	*−2.0*
Carbohydrate metabolism	6-P-beta-glucosidase	spr0276 (*celA*)	*−3.0*
Transcription factors	Antiterminator	spr0279 (*bglG*)	*−3.2*
Transport: Carbohydrates	PTS sugar-specific EII component	spr0280 (*celC*)	*−2.3*
Hypothetical proteins		spr0281	*−2.3*
Transcription factors	Antiterminator	spr0504	*−2.2*
Pathogenesis	LPxTG protein	spr0561 (*prtA)*	**2.3**
Hypothetical proteins		spr0600	*−2.0*
Hypothetical proteins		spr0601	*−2.1*
Hypothetical proteins		**spr0929 (*staR*)**	**438.1**
Transport: antibiotics	ABCT ATP-binding/membrane	spr1289	*−2.2*
Hypothetical proteins		spr1402	**2.3**
Transport: cations	ABCT ATP-binding-Mn	spr1492 (*psaB)*	**4.1**
Transport: cations	ABCT membrane permease -Mn	spr1493 (*psaC)*	**4.7**
Transport: cations	ABCT substrate-binding-Mn	spr1494 (*psaA)*	**4.8**
Carbohydrate metabolism	Galactose-1-P uridylyltransferase	spr1667	**2.1**
Carbohydrate metabolism	Galactokinase	spr1668 (*galK)*	**2.3**
Carbohydrate metabolism	Alcohol dehydrogenase	spr1670 (*adhB*)	**73.7**
Hypothetical proteins		spr1671	**105.3**
Transport: cations	Cation diffusion facilitator	spr1672 (*czcD)*	**507.5**
Carbohydrate metabolism	Dextran glucosidase	spr1698	*−3.2*
Hypothetical proteins		spr1940	*−2.2*
Hypothetical proteins		spr1966	**2.6**
Transport: other	Glycerol uptake facilitator protein	spr1988 (*glpF)*	*−2.2*
Carbohydrate metabolism	Glycerol-3-P dehydrogenase	spr1989	*−2.6*
Carbohydrate metabolism	Glycerol-3-P dehydrogenase	spr1990	*−3.1*
Carbohydrate metabolism	Glycerol kinase	spr1991 (*glpK)*	*−2.0*
Hypothetical proteins		spr2037	**3.1**
Hypothetical proteins		spr2038	**3.4**
Hypothetical proteins		spr2039	**3.4**
Hypothetical proteins		spr2040	**2.0**

^a^ Genes shadowed in gray are those known to be regulated by high ZnSO_4_. ^b^ The responsive genes from two replicates included those showing a significant fold change (absolute value ≥ 2) and a *p*-value-adjusted ≤ 0.01.

## Data Availability

The data discussed in this publication have been deposited in NCBI’s Gene Expression Omnibus and are accessible through GEO Series accession number GSE225261 (https://www.ncbi.nlm.nih.gov/geo/query/acc.cgi?acc=GSE225261).

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
