# Peer review of "StaR Is a Positive Regulator of Topoisomerase I Activity Involved in Supercoiling Maintenance in Streptococcus pneumoniae"

_ijms, 2023, doi:10.3390/ijms24065973_

Round 1
Reviewer 1 Report
GENERAL COMMENTS
This is a nice characterization of a newly discovered protein that interacts with bacterial topoisomerase I of Streptococcus pneumoniae. The English usage is generally good, although another round of editing by a native English speaker would solidify the presentation. Overall, this is a solid paper that continues the excellent work by this laboratory concerning S. pneumoniae chromosome biology. Below are a few specific comments that I hope are useful during revision.
SPECIFIC COMMENTS
Line 2 Can you indicate in the title whether this is a positive or negative regulator? Some journals resist the use of “new, first observation, novel”
Line 16. No hyphen after DNA, yes hyphen after nucleotide
Line 19 in the presence of inhibitor: for the logic to be clear, was this subinhibitory concentrations so you could measure growth? Also, comma before and
Line 22 what does their refer to?
Line 41 logic. While the enzyme may cleave one strand (as opposed to two), where in the cell do you find single-stand DNA except in D-loops or R-loops? You need to be clear with your meaning here.
Line 43 you cited Streptococcus refs, which are from the present lab, but all of this was known in 1982 from E. coli work. You do not want to leave the impression that you are biasing the citations to your own work because that undermines author credibility. See line 49 for balanced citation.
Line 71 comma after topoisomerases
Line 88 from?
Line 91 gyrase inhibitor. Is the key idea that novo relaxes chromosomal DNA or that it inhibits gyrase, in which case why not use a quinolone? (oxo does not relax DNA at MIC; why is that the case if topo I is responsible for relaxation?). I would shift emphasis to relaxation rather than gyrase inhibition.
Line 127 deepen – awkward
Line 142 what about a “Thus” sentence here? The reader needs to consolidate the detail in this paragraph before moving on.
Line 200 in vivo. The measurements were made in vitro, and the presence of wrapping proteins in vivo could have affected the real level of supercoiling in vivo. See Sinden and Pettijohn 1980 for in vivo measurements. You need to rephrase the header and the text to keep the logic clean. Also, plasmid supercoiling does not always reflect chromosomal supercoiling. You need to show that you are aware of these issues and not overstate the meaning of the evidence..
Line 205 the value. The general reader will see such a broad distribution of topoisomers that they will wonder how you could get a single number that is meaningful. You should address this. I’m not sure that average linking number is the correct term. What would happen if you placed the comparative gels next to each other?
Line 211 role. You can help the reader by making this a more precise term to indicate the direction of the role
Line 244 nucleoid not nucleoide
What about repeating fig 6E to see sharp bands of topoisomers?
Line 336 change
Line 367 inhibition of gyrase. Do you believe that relaxation from novo is due to inhibition of gyrase plus action of topo I? This is the interpretation of what you write. But what if novo causes gyrase to work backwards? Take a look at old E. coli data. Does co-treatment with nalidixic and coumermycin block relaxation? Does novo or cou cause gyrase to relax plasmid DNA in vitro? In the overall picture this is not important, but word choice and phrasing shows how well authors know the literature.
Line 378 down regulation: only if the proteins turn over?
Line 395 50% association. Remind us what the association percent is for HU as a comparator. Is 50% a convincing number? Some readers might see this as random distribution.
Line 403 identity? What do you mean? Something about identical surfaces interacting?
Line 409 ascertain: probably the wrong word
Line 419 interact with
Reviewer 2 Report
This ms describes the characterization of a novel protein, StaR, thought to be involved in supercoiling maintenance in Streptococcus pneumoniae. StaR is a hypothetical protein that has low-level similarity to the E. coli nucleotide-associated protein (NAP) YejK, and is characterized on this basis. Treatment of Streptococcus pneumoniae bacteria deleted for the staR gene with novobiocin suggests that it may have a role in supercoiling control. The protein (StaR) was expressed in E. coli enabling in vitro experiments to be performed. Although these observations are of potential interest, I found there to be a number of problems with the work that may preclude publication at this point.
Major issues:
1. P. 4 and Fig. 2: this experiment is confusing and hard to follow. The assertion that overproduction of StaR affects novobiocin susceptibility does not seem to be clear from the data.
2. Fig. 4: the differences in supercoiling +/- Zn seem very slight and I really wonder if they are significant.
3. Fig. 5: this is confusing. I think there should be better evidence that StaR localises to the nucleoid. Panel B would also benefit from an inset like panel A.
4. Fig. 6 is very unconvincing. Panels A and C suggest that a huge excess of StaR over topo I (~1000-fold) is needed to see an effect. In panel B the nicked circular species (OC) is quantitated, which is presumably due to non-specific nuclease activity; they should quantitate the relaxed topoisomers. The quality of the supercoiling assay in panel E is very poor.
Minor issues:
i. The English needs attention throughout.
ii. P. 7, lines 204-212: the authors refer to Fig. 5 when I think they mean Fig. 4.
Round 2
Reviewer 2 Report
This is a revised version of a ms describing the characterization of a novel protein, StaR, thought to be involved in supercoiling maintenance in Streptococcus pneumoniae. The authors have responded to my original comments; I give below my responses to their rebuttal:
Major issues:
1. P. 4 and Fig. 2: this experiment is confusing and hard to follow. The assertion that overproduction of StaR affects novobiocin susceptibility does not seem to be clear from the data. The authors’ response is satisfactory.
2. Fig. 4: the differences in supercoiling +/- Zn seem very slight and I really wonder if they are significant. I still feel that the differences are small, but the authors have attempted to address this point.
3. Fig. 5: this is confusing. I think there should be better evidence that StaR localises to the nucleoid. Panel B would also benefit from an inset like panel A. The authors’ response is satisfactory.
4. Fig. 6 is very unconvincing. Panels A and C suggest that a huge excess of StaR over topo I (~1000-fold) is needed to see an effect. In panel B the nicked circular species (OC) is quantitated, which is presumably due to non-specific nuclease activity; they should quantitate the relaxed topoisomers. The quality of the supercoiling assay in panel E is very poor. I still feel that this experiment is problematic:
a. Although the method of quantitation has improved, it is still puzzling that such a large excess of StaR is needed to see an effect.
b. Did the authors consider testing the effect of StaR on topo IV?
c. In the new panel E is looks to me at though supercoiling is being stimulated by StaR; did the authors quantitate the sc band as a % of the total?
d. In panel D they ought to put the StaR concentrations above the gel lanes.
Additional points:
i. In Fig. 7 it is worrying tht the topo I antibody does not pull down StaR in the presence of Zn. This would be more convincing if they had also done gyrase pull downs.
ii. Minor point: in fig. 3B StaR is mis-labelled.
Overall the ms is improved but I still have concerns about certain aspects, particularly Fig. 6; these effects could just be non-specific protein effects.
Minor issues have been attended to.
Author Response
We thank the referee for his/her comments. We appreciate his/her positive response to most of our rebuttal.
We have revised the manuscript attending the new reviewer comments, making the changes required and paying special attention to give a positive response to the issues raised.
- Fig. 6 is very unconvincing. Panels A and C suggest that a huge excess of StaR over topo I (~1000-fold) is needed to see an effect. In panel B the nicked circular species (OC) is quantitated, which is presumably due to non-specific nuclease activity; they should quantitate the relaxed topoisomers.
The quality of the supercoiling assay in panel E is very poor. I still feel that this experiment is problematic:
- Although the method of quantitation has improved, it is still puzzling that such a large excess of StaR is needed to see an effect.
Although there is a large excess of StaR (2-4 µM StaR versus 1nM of TopoI) in the reactions, this protein activates TopoI, while has not effect on gyrase (4 nM of gyrase) activity at the same concentration. At even higher concentrations (4-16 µM StaR) the protein does not relax CCC plasmid. Therefore, we believe that StaR is indeed an activator of TopoI.
- Did the authors consider testing the effect of StaR on topo IV?
In S. pneumoniae Sc is mostly controlled by the opposing activities of Topo I and gyrase. Therefore, since the effect on Sc was compatible with either activation of TopoI or inactivation of gyrase, we do not consider testing Topo IV activity in the presence of StaR.
- In the new panel E is looks to me at though supercoiling is being stimulated by StaR; did the authors quantitate the sc band as a % of the total?
Gyrase activity was quantified and the results were already included in the revised version of the manuscript: “In contrast, no activation of S. pneumoniae DNA gyrase activity was observed, since 30% of CCC form was observed at all StaR concentrations tested”. These data derived from the quantification of two additional experiments performed in the first revision.
However, we have now included data from the experiment showed in the first version of the manuscript. Now, a total of three experiments of Gyrase activity in the presence of StaR have been considered for quantification. As suggested previously by this referee, the percentage of topoisomers+ CCC has been considered for gyrase activity estimation. Results showed that Gyrase activity was not affected by StaR, and maintained at 65% at all StaR concentrations tested. A new panel (F) with activity quantification data is now included on Figure 6. Accordingly, the sentence of the revised version 1: “In contrast, no activation of S. pneumoniae DNA gyrase activity was observed, since 30% of CCC form was observed at all StaR concentrations tested” has now been changed to: “In contrast, no activation of S. pneumoniae DNA gyrase activity was observed, since 73% of CCC plus topoisomers was observed at all StaR concentrations tested.
- In panel D they ought to put the StaR concentrations above the gel lanes.
Thank you for this observation, we apologize for this mistake. StaR concentrations are now indicated in panel D.
Additional points:
- In Fig. 7 it is worrying tht the topo I antibody does not pull down StaR in the presence of Zn. This would be more convincing if they had also done gyrase pull downs.
We agree with this observation, we do not have an explanation for that, as it is reflected in the manuscript “However, no pulldown of StaR by TopoI-Ab was observed, suggesting that the polyclonal antibody against TopoI dissociates the TopoI-StaR complex”.
Given that StaR has not effect on gyrase activity, we did not consider to perform gyrase pull-down experiments.
- Minor point: in fig. 3B StaR is mis-labelled. OK, thank you, it has been changed.
Overall the ms is improved but I still have concerns about certain aspects, particularly Fig. 6; these effects could just be non-specific protein effects.
Minor issues have been attended to.